# The Role of Laser Photocoagulation in Treating Diabetic Macular Edema in the Era of Intravitreal Drug Administration: A Descriptive Review

**DOI:** 10.3390/medicina59071319

**Published:** 2023-07-17

**Authors:** Miho Nozaki, Ryota Ando, Toshiya Kimura, Fusae Kato, Tsutomu Yasukawa

**Affiliations:** 1Department of Ophthalmology, Laser Eye Center, Nagoya City University East Medical Center, Nagoya 464-8547, Japan; 2Department of Ophthalmology and Visual Science, Nagoya City University Graduate School of Medical Sciences, Nagoya 467-8601, Japan; 3Department of Ophthalmology, Toyota Kosei Hospital, Toyota 470-0396, Japan

**Keywords:** focal laser, grid laser, navigated laser, pattern scan laser, subthreshold laser, selective retina laser, microaneurysm, endpoint management laser, indocyanine-green angiography, optical coherence tomography (OCT)

## Abstract

*Background and Objectives*: This study aimed to elucidate the role of laser photocoagulation therapy in the treatment of diabetic macular edema (DME) as an alternative to, or in conjunction with, the first-line treatment, anti-vascular endothelial growth factor (VEGF). *Materials and Methods*: A comprehensive literature search to identify studies that evaluated the efficacy of laser photocoagulation therapy in the management of DME was performed. The relevant findings of the efficacy of focal/grid laser therapy from data in randomized, controlled trials were synthesized, and the potential of new laser technologies, such as navigated laser systems, pattern scan lasers, and subthreshold lasers, was explored. The usefulness of multimodal imaging-guided laser therapy was also evaluated, with a focus on the potential contribution to anti-VEGF therapy. *Results*: Focal laser photocoagulation targeting microaneurysms remains an effective therapeutic approach to chronic refractory edema, despite the widespread use of anti-VEGF therapy. To achieve the best possible treatment outcomes, precise identification of microaneurysms is essential. This requires the use of multimodal imaging-guided, highly accurate, minimally invasive coagulation techniques. Subthreshold laser therapy can also reduce the frequency of anti-VEGF injections and minimize treatment burden. *Conclusions*: Further studies are needed to determine the optimal timing and settings for laser photocoagulation therapy and the potential of new laser technologies in the management of DME. Nevertheless, laser photocoagulation therapy plays an important role in the management of DME, in conjunction with anti-VEGF therapy.

## 1. Introduction

Diabetic macular edema (DME), rather than proliferative diabetic retinopathy, is an increasingly common cause of visual impairment [1]. Macular laser photocoagulation used to be the primary treatment option for DME since The Early Treatment Diabetic Retinopathy Study Research Group (ETDRS) showed that focal/grid laser therapy could stabilize visual acuity [2]. However, with the introduction of anti-vascular endothelial growth factor (VEGF) drugs, the use of macular laser photocoagulation has decreased significantly [3,4,5]. The EURETINA guideline notes that laser treatment is no longer considered the standard of care for DME in the current era of anti-VEGF drugs [6]. Whereas anti-VEGF therapy has proven to be effective in DME [7,8,9,10], real-world data have shown that relying solely on anti-VEGF injections may not be practical due to factors such as the economic burden of treatment, poor response to anti-VEGF, and systemic conditions that limit treatment options [11,12,13].

Fortunately, new technologies in laser photocoagulation have been developed to address these challenges. These technologies include navigated laser systems [14], subthreshold laser therapies [15,16], and pattern scan laser systems [17]. In addition, advances in multimodal imaging technology have had a transformative impact on laser photocoagulation procedures, enabling more accurate, targeted, minimally invasive treatment options for patients. These advances have led to renewed interest in the use of laser photocoagulation as a treatment for DME.

In this review, the new techniques and indications for laser therapy in the treatment of DME in the anti-VEGF era are discussed. The potential benefits of incorporating these new laser technologies into the treatment plan, and the new role that laser photocoagulation may play in the management of DME are explored. By considering these new developments in laser technology, it may be possible to improve treatment outcomes and offer more options for patients with DME.

## 2. Focal/Grid Laser Therapy

### 2.1. ETDRS

The Early Treatment Diabetic Retinopathy Study (ETDRS) used fluorescein angio- grams to classify DME eyes with focal leakage associated with microaneurysms as focal and DME eyes with less leakage associated with microaneurysms as diffuse [2,18].

In the ETDRS, focal laser photocoagulation was directly applied to microaneurysms to obtain closure of the leak, and grid laser photocoagulation was applied to areas of diffuse leakage or thickened retinae [2,18,19]; these techniques became the gold standard for treating DME. The ETDRS suggested that focal/grid laser could prevent severe vision loss compared to observation only [2]. However, focal/grid laser therapy might result in scar atrophy and subretinal fibrosis in the macula in approximately 21% of patients, leading to vision loss (Figure 1) [20]. To avoid these complications, DRCR.net (Diabetic Retinopathy Clinical Research Network) recommended “modified” ETDRS settings, in which the laser spots were slightly smaller (50 μm) and lighter (gray), and a color change was not required when treating the microaneurysms [21]. DRCR.net protocol A found the modified ETDRS laser to be better than the Mild Macular Grid laser for DME in terms of optical coherence tomography (OCT)-based retinal thickness measures, but best-corrected visual acuity (BCVA) measures in the two groups were not substantially different [21]. In Protocol A, researchers specifically evaluated the effect of modified ETDRS laser in eyes with non–center-involved clinically significant macular edema (CSME). Vision was stable, and the study concluded that modified ETDRS focal/grid laser therapy was still an appropriate treatment for extrafoveal DME [22]. The EURETINA guideline also recommended laser application as a treatment option, specifically targeting the vasogenic subform of DME, characterized by the presence of focal accumulation of microaneurysms and leaking capillaries [6].

### 2.2. DRCR Network

Protocol I conducted by DRCR.net compared the efficacy of ranibizumab with prompt or deferred focal/grid laser therapy in treating DME. Prompt laser therapy was defined as that within 3 to 10 days of the diagnosis or injection. Deferred laser therapy meant that performed more than 6 months after the initial injection. This study demonstrated the superiority of anti-VEGF treatment over focal/grid laser therapy [7,8]. Furthermore, in Protocol I, it was observed that, when the baseline visual acuity was worse than 20/50, the deferred laser group demonstrated superior BCVA at the 5-year follow-up compared to the prompt laser group [23]. Since 56% of the deferred laser group had never received focal/grid laser therapy in the 5 years, deferring focal/grid laser therapy may be associated with a greater chance of relatively greater improvements in visual acuity over 5 years compared with adding focal/grid laser therapy when initiating intravitreous ranibizumab, especially for eyes with worse visual acuity at baseline.

### 2.3. Can Focal/Grid Laser Therapy Reduce the Number of Anti-VEGF Injections?

The efficacy of anti-VEGF therapy has been well established in multiple randomized, controlled trials (RCTs), such as RISE/RIDE [24], RESTORE [10], and DA VINCI [25], which have all shown its superiority over laser photocoagulation. Recently approved anti-VEGF drugs have been compared to aflibercept in clinical studies [26,27], and no further studies have been conducted to compare the efficacy of focal/grid laser treatment alone.

Nonetheless, administering frequent anti-VEGF injections in line with the protocols used in RCTs poses a significant challenge in real-world clinical practice [11,12,13]. The combination therapy of laser photocoagulation and anti-VEGF drugs may have potential in reducing the number of injections needed for treatment. The READ-2 study compared the efficacy of ranibizumab (0.5 mg) with/without laser, and they found that combination therapy with focal/grid laser treatment might reduce the percentage of patients that require frequent ranibizumab injections. The number of injections in 2 years was 9.3 in the monotherapy group and 2.9 in the combination therapy group. However, they also found that the monotherapy group did not have patients with resolved or controlled edema who had poor visual acuity, whereas this was the case in 22% of combination therapy patients. These indicated that adding focal/grid laser therapy might sacrifice the visual outcome [9].

In Protocol I, the mean change in the visual acuity letter score from baseline to the 5-year follow-up tended to be superior in the ranibizumab with deferred laser group than in the ranibizumab with prompt laser group, with +9.8 letters versus +7.2 letters, respectively (*p* = 0.09). The median number of injections was 17 versus 13 in the deferred and prompt laser groups, respectively [23]. Although the RESTORE study showed the effectiveness of combining laser photocoagulation with ranibizumab (0.5 mg), there was no significant difference in the frequency of injections required [10]. Further studies will need to identify the best timing of adding laser photocoagulation to provide better anatomical and functional improvement and reduce the number of injections.

### 2.4. Steroid vs. Focal/Grid Laser

There is strong evidence supporting the involvement of chronic inflammation in the pathophysiology of DME, highlighting the beneficial effects of corticosteroids in improving DME [7]. However, in the current era of anti-VEGF treatment, the use of steroids as a therapeutic option for DME is considered second-line therapy. This is primarily due to the potential complications associated with steroids, including cataract formation and elevated intraocular pressure [7]. The EURETINA guidelines recommend considering the use of steroids as the first choice for patients with a history of a major cardiovascular event, since these patients were excluded from major anti-VEGF trials [6]. In addition, corticosteroids may be considered a first-line therapy option for patients who are unwilling to undergo monthly injections and monitoring during the initial 6 months of treatment [6]. Furthermore, due to cataract progression, pseudophakic patients are well-suited for steroid therapy [6,7]. The PLACID trial reported the results of a comparison of a dexamethasone implant (Ozurdex) with laser treatment versus laser monotherapy in patients with DME. Though there was no difference between the groups at 1 year, the percentage of patients gaining at least 10 letters was significantly higher in the Ozurdex plus laser group than in the laser monotherapy group at months 1 and 9 [28]. Although no specific study has directly compared fluocinolone acetonide (Iluvien) with laser treatment versus laser monotherapy, available evidence suggests that steroid therapy, in combination with laser treatment, appears to be superior to laser monotherapy [8,28].

## 3. New Laser Technologies

### 3.1. Navigated Laser Systems

The navigated laser photocoagulator (NAVILAS; OD-OS GmbH, Teltow, Germany) is a novel laser technology that incorporates a retinal eye-tracking laser delivery system and integrated digital fundus imaging. This system features advanced image processing to help treat retinal lesions by overlaying images onto the live image of the retina [14]. It was designed to improve accuracy and make it easier to treat specific lesions such as microaneurysms that require precise treatment [29]. Physicians create a treatment plan using fundus photography, fluorescein/indocyanine green angiography, or OCT, which is then displayed on a real-time image of the patient’s fundus during treatment to guide the process. By using NAVILAS, which incorporates an eye-tracking system and pre-registered laser planning, the hit rate for microaneurysms increased by 30% compared to the conventional slit-lamp delivery system [30]. In the study that compared the efficacy of the NAVILAS macular laser versus the conventional slit-lamp-based macular laser for treating DME, the NAVILAS group showed significantly better BCVA than the conventional laser group after 3 months. In addition, within the first 8 months, the NAVILAS group had a lower re-treatment rate (18%) compared to the conventional laser group (31%). These findings suggest that the NAVILAS macular laser may be a more effective treatment option for DME than the conventional slit-lamp-based macular laser [31]. Liegl et al. reported that the burden of anti-VEGF treatment for DME can be significantly reduced by the use of a precise focal laser with the NAVILAS laser system [32]. They found that NAVILAS laser combination therapy and ranibizumab monotherapy had similar effects on the mean BCVA letter score (8.41 vs. 6.31 letters, *p* = 0.258), but the combination group required significantly fewer injections after the loading phase (0.88 ± 1.23 vs. 3.88 ± 2.32, *p* ≤ 0.001). By month 12, 84% of patients in the monotherapy group needed additional ranibizumab injections, compared to 35% in the combination group (*p* ≤ 0.001) [31]. They also found that the effect lasted for 36 months. Patients who received the combination treatment required significantly fewer injections than those who received only anti-VEGF. Specifically, they needed two fewer injections in year 1 and 1.3-times fewer injections in years 2 and 3 (N = 24) [33]. However, in the TREX-DME study, which compared the long-term effects of treat-and-extend dosing of ranibizumab with and without navigated focal laser therapy for DME, there was no significant difference in BCVA or the number of ranibizumab injections needed (N = 109) [34]. Kato et al. reported that focal laser photocoagulation with the NAVILAS + 577 (wavelength 577 nm) was effective in treating refractory DME by reducing central retinal thickness and macular edema after 6 months. In their study, 11 eyes received an injection of aflibercept following initial navigated laser, and their number of injections for 6 months was 1.55 ± 1.0; for comparison, 23 eyes in their institution that were treated with anti-VEGF monotherapy received an average of 2.52 ± 1.0 injections during the same 6-month period [35]. More recently, an economic modeling study demonstrated that using the NAVILAS laser decreased patient injection burden while achieving comparable clinical outcomes [36]. Further studies involving a larger patient population are needed to determine whether the combination of precise focal laser photocoagulation using the NAVILAS laser is effective in reducing the number of anti-VEGF injections required.

Several studies have demonstrated the benefits of using the NAVILAS focal laser for treating refractory DME compared to other treatments [35,37]. The prospective study that treated 8 eyes with refractory DME showed that focal laser photocoagulation using NAVILAS significantly reduced central retinal thickness and macular volume, with a significant improvement in BCVA [36]. Hirano et al. reported that eyes with perifoveal leaking microaneurysms required a larger number of aflibercept injections, despite being treated with a combination of focal/grid laser therapy and aflibercept injections [38]. This is likely due to the limitation of conventional focal/grid laser treatment, which can only be applied outside the fovea (>500 μm). In that study, they demonstrated the efficacy of the NAVILAS focal laser for refractory DME; the mean distance from the center of the fovea to the closest microaneurysms was 624.8 ± 377.7 μm [37], and 63% had microaneurysms located at the perifovea, within less than 500 μm from the center of the fovea [37]. Based on the evidence presented, it can be inferred that conventional focal/grid laser treatment may have limited efficacy in treating refractory DME due to the proximity of microaneurysms to the fovea. Focal laser treatment using the NAVILAS laser may be a more suitable option for treating refractory DME due to its precise and accurate approach.

An additional advantage of the NAVILAS laser is the ability to plan laser treatment based on images obtained from multimodal imaging. This innovative feature enables more precise and customized treatment plans, tailored to the specific needs of each patient. The role of this technology will be discussed in greater detail in Section 4.

### 3.2. Subthreshold Laser

Conventional laser photocoagulation therapy poses a risk of RPE atrophy, which can lead to central scotoma or scar enlargement [20,39]. However, subthreshold micropulse laser (SMPL) therapy has been developed as a safer alternative, delivering laser energy below the threshold required for permanent tissue destruction. It was found that, by shortening the irradiation time, the RPE could be selectively coagulated [40]. The principle of subthreshold laser photocoagulation was then developed by controlling the on/off time of irradiation to achieve very short irradiation times only to the RPE. Friberg and Karatza first reported the usefulness of subthreshold laser treatment for macular edema using a micropulsed subthreshold laser with a diode laser wavelength of 810 nm [41]. Since then, numerous clinical studies have investigated the use of SMPL for the treatment of DME [42,43,44,45,46,47], and several reports have shown that combination therapy with SMPL might be effective to reduce the number of anti-VEGF injections in the treatment of DME [42,43,44,45,46]. Mansouri et al. [48] reported that SMPL was more effective in DME patients with central macular thickness (CMT) less than 400 μm. Other previous prospective studies also showed that SMPL was effective when used with intravitreal injections of aflibercept with the pretreatment CMT below 400 μm [44,46] or 450 μm [45] after at least 3 loading doses of aflibercept. Based on this evidence, initial anti-VEGF treatment with deferred SMPL laser therapy could be useful in reducing the number of anti-VEGF injections. According to Luttrull et al., transfoveal SMPL therapy has been shown to be safe and effective in patients with good visual acuity [49]. Their study also found that SMPL treatment yielded positive outcomes in eyes with CMT below 300 μm [49]. SMPL laser treatment could be considered a suitable option for DME patients with good preoperative visual acuity who are not suitable candidates for conventional photocoagulation or intravitreal injection [50]. The EURETINA guideline also recommends SMPL for eyes with higher visual acuity affected by early diffuse DME, since it provides a cost-effective alternative that helps avoid collateral thermal diffusion and associated chorioretinal damage [6].

However, the variability in the use of wavelengths, laser powers, and duty cycles of SMPL lasers presents a challenge in comparing outcomes across clinical trials. Therefore, establishing optimal laser parameters and standardizing SMPL laser settings are critical steps towards achieving wider acceptance and integration of SMPL lasers into clinical practice.

Other laser devices are also equipped with a subthreshold laser mode (Table 1). The NAVILAS laser has a “Microsecond Pulsing Mode” that controls the irradiation time for subthreshold laser coagulation, similar to the micropulse laser (Figure 2) [51]. Since the subthreshold laser is designed to be “invisible,” using the NAVILAS system to retain laser location information and settings may provide a significant advantage.

Selective Retina Laser Therapy (SRT) is another new subthreshold laser [52]. Unlike conventional laser photocoagulation therapy, SRT’s mechanism of action involves the generation of microbubbles around melanosomes in the RPE, which rupture and damage only the RPE while leaving the surrounding neural retina or choroid unaffected [53,54]. Subthreshold lasers do not produce visible laser spots, which can make it difficult to ensure effective treatment and increase the possibility of under-treatment. However, SRT uses optoacoustic techniques to provide real-time feedback on the degree of RPE damage, which overcomes this limitation. With this feedback, the treating physician can avoid both under-treatment and over-treatment, ensuring optimal treatment outcomes [55]. Several groups have reported the efficacy of SRT in treating DME [56,57,58]. It is interesting to note that, similar to the SMPL, SRT resulted in significant improvement in visual acuity and retinal thickness in patients with pre-treatment central foveal retinal thickness less than 400 μm, and there was almost no need for additional anti-VEGF drug injections in this group. However, in patients with pre-treatment retinal thickness of 400 μm or greater, there was no significant improvement in visual acuity or retinal thickness, and 73% of these patients required rescue treatment with anti-VEGF drugs [58]. Therefore, performing SRT as a deferred laser therapy after initiating anti-VEGF drug treatment may be useful. 2RT is an innovative nanopulse laser technology that delivers only around 0.2% of the energy per pulse compared to standard photocoagulation techniques [59]. With 2RT laser treatment, the effects are localized within the RPE, ensuring minimal impact on surrounding retinal tissues. A pilot, randomized trial evaluating the use of 2RT in patients with DME demonstrated its non-inferiority compared to conventional laser treatment [59]. Further studies are warranted to explore the efficacy and safety of 2RT in a larger population. 

### 3.3. Pattern Scan Laser

The PASCAL^®^ (Topcon Medical Laser systems, Santa Clara, CA, USA) is a semi-automated laser system that uses pattern scanning technology to deliver multiple laser burns in a single application [17]. This technology allows for precise and controlled delivery of arrays with predetermined parameters, resulting in reduced procedural time. To deliver multiple laser burns in a single application, the duration of laser photocoagulation with the PASCAL laser is set to 0.02 s, which is much shorter than the conventional laser setting of 0.2 s. This short-pulse setting offers several advantages over conventional laser treatment. Short-pulse laser treatment is faster, generates less heat, and causes less discomfort to the eyes [17,60]. Furthermore, research suggests that short-pulse laser treatment may induce less inflammation, fewer inflammatory cytokines in the sensory retina, and less macular thickening in patients with diabetic retinopathy compared to use of the conventional pulse duration [61,62]. Despite the advantages of short-pulse laser treatment, studies suggest that it may be less effective than conventional laser treatment for high-risk proliferative diabetic retinopathy. These studies propose that the total area of panretinal photocoagulation (PRP) scars generated by the conventional laser exceeds that of the short-pulse laser, despite both groups receiving the same number of laser spots [63]. This difference may be due to the tendency of photocoagulation scars generated by the conventional laser to expand after treatment [64]. However, other reports suggest that the expansion rate of photocoagulation scars generated by the short-pulse laser is lower than that of the conventional laser [65]. Conversely, the limited scar expansion observed with short-pulse lasers has led to the development of new modified settings for focal/grid lasers. In contrast to the “modified” ETDRS technique, which has been widely used for years, these newly modified settings now use shorter pulse durations, such as 0.02–0.03 s [35,37,66,67,68].

Endpoint Management (EpM) is another subthreshold laser method provided by the short-pulse pattern scan laser system PASCAL [16]. The EpM laser uses a continuous wave, and the EpM algorithm uses the Arrhenius integral formula to adjust the laser power and pulse duration, enabling the achievement of therapeutic effects while minimizing damage to surrounding tissues [16,69]. There have been several reports that the EpM laser was safe and effective for DME treatment. Hamada et al. reported that EpM laser monotherapy was effective in reducing central macular thickness, but visual acuity, macular sensitivity, and macular volume were not significantly improved (N = 10) [70]. The setting that this study used was 50% of the threshold (size, 100 μm; duration, 0.015 s; spacing, 0.5; and energy, 4.5–7.8 mJ). Several prospective studies aimed to investigate whether adding the EpM laser to anti-VEGF drug therapy could reduce the number of anti-VEGF injections required to treat DME. Though one study, the END-DME study, demonstrated that combination therapy with the EpM laser could reduce the number of anti-VEGF injections [71], another study did not find such an effect [72]. Both studies, however, used the same EpM threshold setting of 40%. Another study compared the efficacy of EpM 30%, EpM 50%, and SMPL (810 nm) for treating DME. The results showed that, though the micropulse system with a 15% duty cycle improved functional outcomes, neither EpM treatment was effective [73].

The effectiveness of EpM lasers for the treatment of DME is still under investigation. Clinical studies have been conducted under various conditions, but the lack of consistency in laser settings and protocols makes it challenging to compare results. Therefore, there is an urgent need to establish optimal laser conditions and indications for EpM therapy to improve the effectiveness and reproducibility of the treatment.

## 4. Multimodal Imaging-Guided Laser Therapy

### 4.1. Indocyanine Green Angiography-Guided Laser Therapy

Fluorescein angiography (FA) has commonly been used to identify microaneurysms in patients with DME. However, several previous reports have suggested the superiority of indocyanine green fundus angiography (ICGA) for detecting the microaneurysms that contribute to the pathogenesis of DME [74,75]. Indocyanine green (ICG) dye binds mostly to serum proteins such as albumin and lipoproteins [76]. Therefore, the dye barely leaks through blood vessels, with the ICGA-positive microaneurysms persisting after the fading of the plasmatic fluorescence on the late-phase ICGA images. The mechanism underlying the superiority of ICGA in detecting microaneurysms that contribute to the pathogenesis of DME may involve the amphiphilic nature of ICG, which enables it to bind to hydrophobic intraluminal materials such as fibrin or lipids. In contrast, fluorescein, being hydrophilic, may be blocked by such intraluminal material. This may explain the preferential detectability of leakage points responsible for macular edema and the enlargement of spots in the late phase of ICGA. In cases of diffuse DME initially diagnosed using FA, ICGA imaging has proven useful in detecting microaneurysms or focal leakage spots [74]. ICGA-guided focal laser therapy has an additional benefit because it leads to significantly fewer targeted microaneurysms than FA-guided focal laser therapy, and ICGA-guided focal laser therapy might be helpful for reducing damage to the retina (Figure 3) [37,74]. Paques et al. identified the relatively large ICGA-positive area as telangiectatic capillaries (defined by a diameter larger than 150 µm on late-phase ICGA) [75,77]. Telangiectatic capillaries may be associated with chronic refractory macular edema [78,79,80], and focal direct laser photocoagulation of telangiectatic capillaries seems to be effective and safe for reducing the burden of anti-VEGF drugs in patients with refractory ME [81].

### 4.2. OCT-Guided Laser

Conventionally, to perform focal/grid laser treatment, dye-based fundus angiography has been necessary to determine the location of the treatment site. However, this procedure can be invasive and uncomfortable for patients. Using OCT instead of dye-based angiography would eliminate the need for an invasive procedure and improve the patient’s comfort during the treatment. OCT thickness maps can provide the topographic location and morphologic pattern of edema, and many physicians use OCT to guide laser photocoagulation, especially for grid laser photocoagulation [82,83]. However, Kozak et al. demonstrated improved accuracy of treatment planning by incorporating information from FA and OCT overlays using the NAVILAS system [84]. Takamura et al. recommended a novel laser photocoagulation protocol, named merged image-guided photocoagulation, for focal direct laser therapy. This protocol involves merging images of the fundus, OCT thickness map, and FA to identify the microaneurysms responsible for macular edema [82]. OCT angiography (OCTA) has emerged as a potential alternative imaging modality for detecting microaneurysms [85]. OCTA-guided focal laser treatment using the NAVILAS system holds promise as a new non-invasive approach for treating microaneurysms responsible for DME [35] (Figure 4). However, the reliability of microaneurysm detection with OCTA varies [86,87,88] and requires further investigation before it can be used to guide focal laser treatment targeting leaky microaneurysms.

OCT can also visualize leaky microaneurysms via B-scan [89,90] as circular structures that encroach upon and disrupt the integrity of the synaptic (inner) portion of the outer plexiform layer (OPL) and induce focal fluid collection, predominantly in the OPL/outer nuclear layer. Furthermore, en face OCT in the deep capillary plexus slab can also identify microaneurysms [91]. OCT-guided focal direct laser photocoagulation showed similar anatomical and functional outcomes compared to conventional laser based on FA [84,85], with significantly less retinal damage [92].

The use of OCT-guided focal laser technique offers several advantages for the management of leaky microaneurysms, including non-invasiveness and real-time imaging capabilities. OCT-guided focal laser therapy may allow precise determination of the proximity of the laser to the microaneurysms, thus increasing the effectiveness of the treatment [93].

### 4.3. Multimodal Imaging Integrated System

The key to performing accurate and minimally invasive laser photocoagulation would be planning laser photocoagulation based on multimodal imaging. Takamura et al. reported the usefulness of merged retinal images, which are obtained from OCT thickness maps and FA and color fundus images, but their technique requires Adobe software (Photoshop CS6 Extended, Adobe Systems Inc., San Jose, CA, USA) to merge the images, and the merged images with the microaneurysms to treat marked were displayed on the computer screen next to the patient and used as a guide while performing laser treatment [85]. Both OCT-guided laser and ICGA-guided focal laser techniques also require a manual step to merge various images onto color fundus images [74,85]. The NAVILAS system is an ideal choice for laser procedures based on multimodal imaging because it allows precise planning of laser treatment by overlaying various digital fundus images on a live fundus image [14].

Recently, PASCAL synthesis (Topcon Medical Laser systems, Santa Clara, CA, USA) developed HUD-1, which is an ophthalmic image projector used as an accessory to PASCAL Synthesis (Figure 5). HUD-1 lets us compare the target area of treatment with a side-by-side view of a reference image previously obtained from the patient’s fundus for location and assessment. With HUD-1, the laser treatment can be performed while maintaining direct visualization of the treatment target area, without requiring the operators to divert their gaze from the slit-lamp microscope.

## 5. The Current Role of Laser Photocoagulation in the Era of Intravitreal Drug Administration

### 5.1. Comparison of Anti-VEGF/Steroid and Laser Treatment

The efficacy of anti-VEGF therapy has been well established in multiple RCTs [10,24,25], all of which have demonstrated its superiority over laser photocoagulation. Clinical studies have recently compared newly approved anti-VEGF drugs to aflibercept [26,27], but no further studies have been conducted to compare the efficacy of focal/grid laser treatment alone. Laser monotherapy is no longer performed as a treatment option for treatment-naive eyes with center-involved DME. It should be noted, however, that current laser photocoagulation techniques encompass various types and settings, such as SMPL, navigation laser, or multimodal imaging-guided laser.

One of the disadvantages of anti-VEGF therapy is the burden of frequent injections and clinic visits. In comparison, laser treatment is generally considered more cost-effective than pharmacotherapy, including both anti-VEGF and steroid treatments [94]. Laser treatment also offers the advantage of a longer-lasting effect compared to anti-VEGF therapy [95]. Moreover, several studies have demonstrated that the combination of laser photocoagulation with anti-VEGF treatment has the potential to reduce the frequency of injections (Table 2) [96]. However, it is important to note that the optimal timing and protocol for combining laser treatment with anti-VEGF injections have not yet been firmly established. Further studies are needed to determine the most effective approach to reduce the treatment burden for patients.

In terms of safety, anti-VEGF injections generally cause fewer topical side effects compared to steroid therapy and laser treatment. Steroid-related side effects may include increased intraocular pressure, cataract formation or progression, and the potential risk of developing steroid-induced glaucoma. On the other hand, advancements in laser treatment have led to the development of current laser settings that aim to minimize side effects [96], such as atrophic creep and the development of scotomata [20]. However, longer-term follow-up studies are still necessary to thoroughly evaluate the absence of side effects of modern retinal laser procedures.

### 5.2. A Real-World Approach: Laser Photocoagulation in Combination with Intravitreal Drug Administration

The types and settings of current laser are summarized in Figure 6. Focal/grid laser photocoagulation is crucial in treating non-center-involved, clinically significant DME [6,22,81]. For center-involved DME, anti-VEGF therapy has become the gold standard treatment. However, despite the benefits seen in RCTs of treating patients more frequently than in real-world practice, the results have shown that approximately 30–40% of patients exhibit a poor response to anti-VEGF treatment [97,98]. The next step would involve considering a switch of intravitreal medication, such as an alternative anti-VEGF agent or steroids [99]. However, if these measures prove unsuccessful, laser therapy could be considered a potential third-line option for managing refractory DME. Whereas microaneurysms in the deep capillary plexus or those detected by ICGA may show resistance to anti-VEGF drugs [37,74,75,76,100], it is generally observed that anti-VEGF therapy effectively reduces microaneurysms [78,101,102]. In addition, steroids have also demonstrated the ability to reduce microaneurysms [103]. Therefore, the current approach involves initiating pharmacotherapy, such as anti-VEGF or steroids, as the initial treatment. Subsequently, deferred focal laser therapy is considered for any remaining microaneurysms that contribute to residual edema, in accordance with the findings from Protocol I [104,105]. In refractory chronic DME cases, ICGA-guided/OCT-guided focal direct laser therapy might be useful for targeting residual microaneurysms. In addition, focal laser therapy using the NAVILAS system may provide benefit in chronic refractory DME cases, because of microaneurysms resistant to anti-VEGF therapy located in the perifoveal vascular network [37,66]. Subthreshold laser techniques have also shown promise in reducing the frequency of anti-VEGF injections for DME treatment. For DME patients with good preoperative visual acuity who are not candidates for intravitreal injection, subthreshold laser treatment could be indicated as a less expensive option [6,49,50].

However, further studies are needed to determine the optimal timing and settings used in these techniques. Nonetheless, laser photocoagulation remains an important tool for treating DME, even in the era of intravitreal drug administration.

## 6. Conclusions

In the future, novel treatments such as gene therapy [106] or anti-inflammatory drugs [99] might be poised to play a crucial role in the management of DME. Nevertheless, recent advancements in laser technology and multimodal imaging have demonstrated new potential applications for laser photocoagulation, even in the era of widespread use of anti-VEGF drugs and steroids. These innovative techniques aim to optimize treatment outcomes while minimizing adverse events and alleviating the burden of frequent injections, thereby offering significant value to DME management. These developments instill hope for patients with DME and underscore the importance of ongoing research in the field of laser photocoagulation.

## Figures and Tables

**Figure 1 medicina-59-01319-f001:**
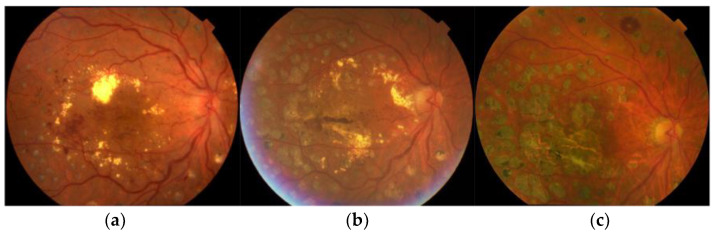
Representative fundus images of atrophic creep after focal/grid laser therapy for DME. This case received focal/grid laser therapy, and (**a**) accumulation and a circinate pattern of hard exudates with numerous microaneurysms are apparent before focal/grid laser therapy. Visual acuity is 20/200; (**b**) Seven years later, macular edema has resolved, and visual acuity is 20/100; (**c**) Fifteen years later, macular edema has resolved completely, but retinal pigment epithelium (RPE) atrophy is noted at the fovea, and the visual acuity has decreased to worse than 20/200.

**Figure 2 medicina-59-01319-f002:**
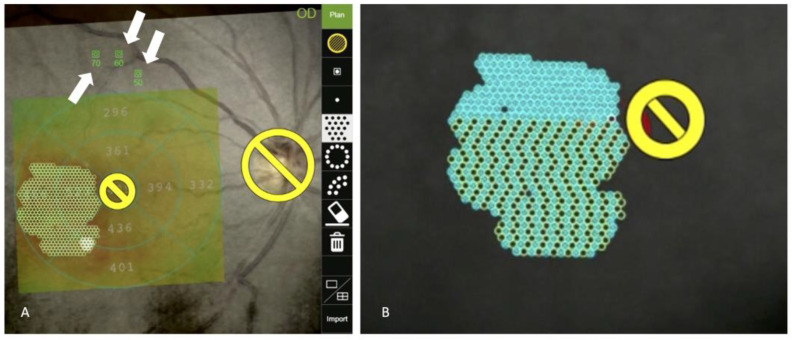
A sample case of subthreshold laser using the NAVILAS microsecond pulsing mode (**A**,**B**). An OCT thickness map is overlaid on a color fundus image. Yellow circles indicate the “no-laser” area, laser planning is set, and titration spots are also planned (white arrows) (**A**). During the procedure, the subthreshold laser may be invisible, but the NAVILAS system can accurately deliver the laser spots as planned with the eye tracking system. This mode delivers the laser in short pulses, alternating among different spots to prevent tissue temperature increase. Blue circles represent spots where the laser treatment has been performed. On the other hand, black circles indicate spots where the laser treatment is yet to be performed.

**Figure 3 medicina-59-01319-f003:**
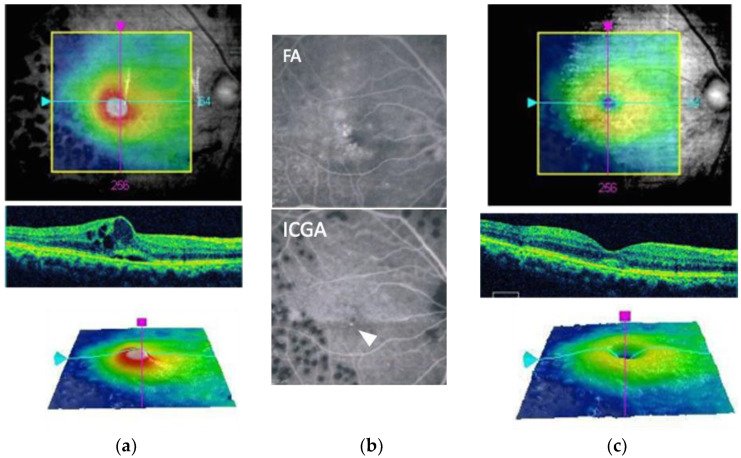
A sample case of Indocyanine Green Angiography (ICGA) -guided NAVILAS focal laser therapy for refractory edema. A 67-year-old man received 4 anti-vascular endothelial growth factor (VEGF) injections, without resolution of edema. (**a**) Before NAVILAS focal laser, subretinal fluid and cystoid macular space are noted in optical coherence tomography (OCT). His visual acuity is 20/25; (**b**) Fluorescein angiography (FA) shows focal leakage from multiple microaneurysms, but ICGA shows one lesion (arrowhead). NAVILAS focal laser therapy is targeted to this ICGA-positive microaneurysm; (**c**) Three months later, OCT shows macular edema has resolved, and his visual acuity remains at 20/25.

**Figure 4 medicina-59-01319-f004:**
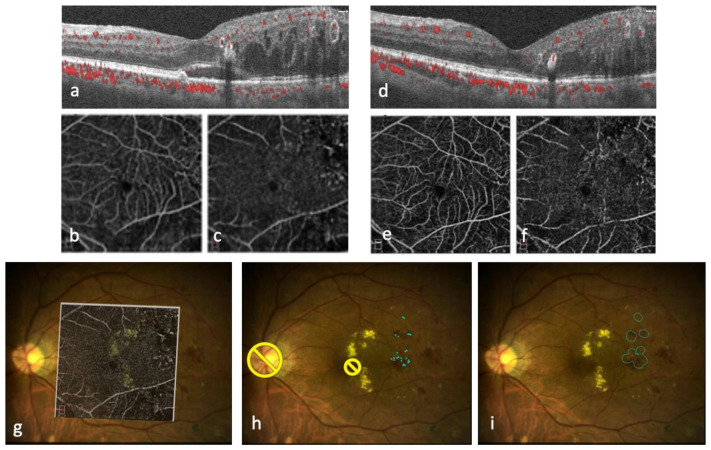
A sample case of Optical Coherence Tomography Angiography (OCTA)-guided NAVILAS focal laser therapy for refractory edema (**a**–**i**). A 54-year-old man received 2 anti-vascular endothelial growth factor (VEGF) injections and conventional focal laser therapy, without resolution of the edema. Before NAVILAS focal laser treatment, subretinal fluid and sponge-like retinal swelling are noted (**a**). His visual acuity is 20/20, and central retinal thickness (CRT) is 405 μm. He has an allergy to fluorescein angiography. Compared to the superficial capillary plexus (SCP) (**b**), more microaneurysms are detected in the deep capillary plexus (DCP) by OCTA (**c**). Four months after OCTA-guided NAVILAS focal laser treatment, there are residual microaneurysms, but subretinal fluid has resolved (**d**). His visual acuity is 20/20, and CRT is 290 μm. The number of microaneurysms is decreased in SCP (**e**) and DCP (**f**). The DCP image (**c**) is overlaid on a color fundus image for planning focal laser using NAVILAS (**g**). Yellow circles indicate the “no-laser” area in the fundus color image. Planned laser spots to individual microaneurysms are seen as blue dots (**h**). Following delivery of focal laser treatment, the treated microaneurysms are visible as faint white dots, delineated by blue circles (**i**).

**Figure 5 medicina-59-01319-f005:**
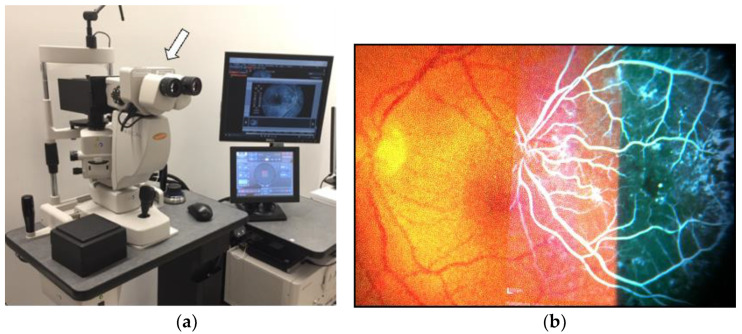
A photo of HUD-1 (arrow), which is mounted on PASCAL Synthesis (**a**); (**b**) physician’s image view with HUD-1. The physician can compare the reference image (fluorescein angiography in this figure) displayed in the binocular microscope view. The reference image can be turned on/off manually.

**Figure 6 medicina-59-01319-f006:**
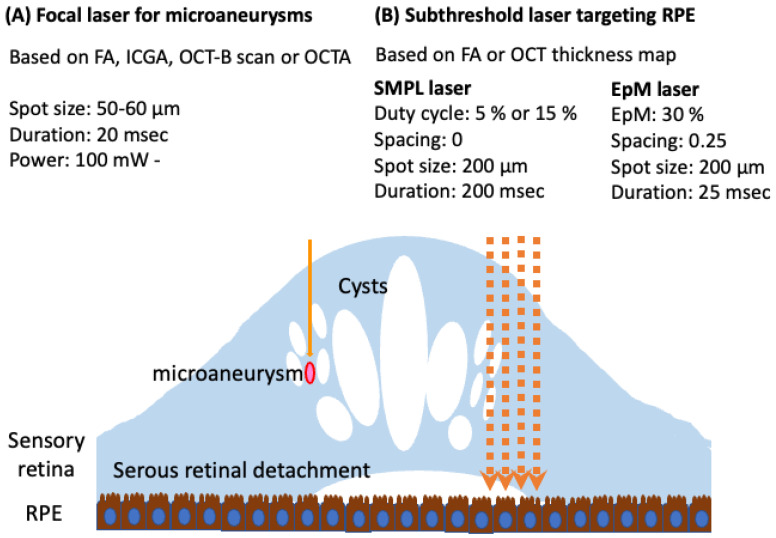
The types and settings of current laser therapies are summarized. (**A**) For focal laser (solid orange arrow) targeting microaneurysms, a short duration of laser application is recommended. For planning laser, the image of fluorescein angiography (FA), indocyanine green angiography (ICGA), optical coherence tomography (OCT) B-scan, or OCT angiography (OCTA) would be used. (**B**) Subthreshold laser (orange dotted arrow) targets the retinal pigment epithelium (RPE). For planning laser, the image of FA or OCT thickness map would be used.

**Table 1 medicina-59-01319-t001:** Types of laser system for diabetic macular edema (DME).

Type of Laser System	Type of Treatment	Advantage	Disadvantage	Device
Navigation laser	Focal laser Microsecond pulsing laser	Accurate image-guided laser with eye tracking	Evaluate laser only via live monitor	Navilas (OD-OS)
Micropulse laser	Micropulse laser	Non-damaging laser with pattern Numerous clinical studies have been reported	Central retinal thickness >400 μm cannot respond	IQ 577, IQ 532 (IRIDEX)
Selective Retina Laser (SRT)	Microsecond laser	Real-time feedback by detecting the pressure wave of microbubbles generated after irradiation	Distribution of the laser device is limited	R: GEN^®^ (Lutronic vision)
Nanosecond laser	Nanosecond laser	Non-damaging laser	One study report in DME	2RT (Ellex)
Pattern Scan (multisport) laser	Focal laser Endpoint management (EpM) laser (PASCAL) SubLuminal laser (EasyRet) Micropulse laser (Supra Scan 577) SmartPulse (Lumenis)	Short-pulse laser shows less expansion of spots	No eye-tracking system Ideal setting has not yet been determined (EpM laser)	PASCAL (IRIDEX/Topcon) EasyRet (Quantel) Supra Scan 577 (Quantel) Smart532 (Lumenis)

DME, diabetic macular edema.

**Table 2 medicina-59-01319-t002:** Summary of studies of combination therapy with anti-VEGF and laser.

First Author, Study Name and Year	Total Eyes	Follow-Up	Intervention	Clinical Results	Number of Injections
Nguyen et al., READ-2, 2010 [9]	IVR = 33, Laser = 33, IVR + laser = 34	24 months	IVR at baseline and months 1, 3, and 5. Focal/grid laser at baseline and month 3 (if needed) IVR and focal/grid laser at baseline and month 3 Starting at month 6, IVR could be given to all groups.	There were no statistically significant differences in BCVA and CMT among 3 groups. However, the monotherapy group did not include patients with resolved or controlled edema who had poor visual acuity, whereas this was the case in 22% of combination therapy patients.	9.3 (IVR monotherapy) 4.4 (laser) 2.9 (combination therapy)
Mitchell et al., RESTORE, 2011 [10]	IVR = 116, IVR + laser = 118, sham + laser = 111	12 months	3 monthly injections followed by as needed. Laser at baseline and as needed.	IVR monotherapy and IVR + laser showed significantly better clinical outcome compared to sham + laser. However, there was no difference between IVR monotherapy and IVR + laser therapy.	7.0 (IVR monotherapy) 6.8 (IVR + laser) 7.3 (laser + sham)
Elman et al., Protocol I 2015 [23]	IVR + deferred laser = 111, IVR + prompt laser = 124	5 years	3 monthly injections followed by as needed. Prompt laser given 7–10 days after initial IVR. Deferred laser given if needed after 6 months	+9.8 letters in IVR + deferred laser and +7.2 letters in IVR + prompt laser (*p* = 0.09) 38% in IVR + deferred laser and 27% in IVR + prompt laser gained at least a 15-letter improvement (*p* = 0.03)	17 (IVR + deferred laser) 14 (IVR + prompt laser)
Liegl et al. 2014 [32]	IVR monotherapy = 32 IVR + Navilas laser = 34	12 months	3 monthly injections followed by as needed. Navilas laser given after 3 loading doses.	Navigated laser combination therapy and IVR monotherapy similarly improved mean BCVA letter score (+8.41 vs. +6.31 letters, *p* = 0.258)	6.88 (IVR) 3.88 (IVR + Navilas laser) (*p* < 0.001)
Payne et al., TREX-DME, 2021 [34]	IVR (0.3 mg) monthly = 24 IVR (0.3 mg) TAE = 40 IVR (0.3 mg) TAE + Navilas laser = 45	3 years	4 monthly injections followed by TAE regimen. Navilas laser given at week 4 and again every 3 months if microaneurysm leakage was present on fluorescein angiography For 3rd year, IVR given as needed.	There were no significant differences among 3 groups.	Third year 3.0 (monthly) 3.1 (IVR TAE) 2.4 (IVR TAE + Navilas laser)
Khattab et al., 2019 [44]	IVA = 27 IVA + SMPL = 27	18 months	3 monthly injections followed by as needed 3 IVAs followed by SPML (within 1 week after the 3rd injection)	Comparable anatomical and visual outcomes between 2 groups.	7.3 (IVA monotherapy) 4.1 (IVA + SMPL) (*p* < 0.005)
Koushan et al., DAM, 2022 [47]	IVA = 15 IVA + SMPL = 15	48 weeks	1 injection followed by as needed SMPL given on the day of the first injection	Eyes that received SMPL showed a numerically greater improvement in BCVA, although this was not statistically significant.	8.5 (IVA monotherapy) 7.9 (IVA + SMPL) (*p* = 0.61)

IVR, intravitreal ranibizumab; BCVA, best corrected visual acuity; CMT, central macular thickness; TAE, treat and extended; IVA, intravitreal aflibercept; SMPL, subthreshold micropulse laser.

## Data Availability

The datasets generated during and/or analyzed during the current study are available from the corresponding author on reasonable request.

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
