# Peer review of "The Role of Laser Photocoagulation in Treating Diabetic Macular Edema in the Era of Intravitreal Drug Administration: A Descriptive Review"

_medicina, 2023, doi:10.3390/medicina59071319_

Round 1

Reviewer 1 Report (Previous Reviewer 2)

Good revision !

English is fine.

Author Response

Thank you very much for valuable suggestion to improve our manuscript.

Reviewer 2 Report (Previous Reviewer 3)

The revised manuscript can now be accepted.

Author Response

Thank you very much.

Reviewer 3 Report (New Reviewer)

1. The source of figures and pictures should be mentioned: The manuscript should provide clear references for all figures and pictures used. It is important to cite the original sources or indicate if the figures are adapted or modified from other studies.

2. Addition of articles and citation discussion: The manuscript would benefit from the inclusion of relevant articles, such as "Laser Therapy in the Treatment of Diabetic Retinopathy and Diabetic Macular Edema" by Everett et al. These articles should be properly cited and discussed within the context of the review to strengthen the arguments and provide a comprehensive analysis of the topic. It would be valuable to consider including more recent studies in the literature review to ensure the manuscript reflects the most up-to-date research in the field. This can enhance the relevance and reliability of the review's findings and conclusions.

3. More information about the combination of photocoagulation therapy with anti-VEGF therapy: The manuscript should provide additional information about the combination of photocoagulation therapy and anti-VEGF therapy in the treatment of diabetic macular edema. This could include discussing the rationale for combining these treatments, their synergistic effects, and any potential challenges or considerations when using them in combination.

4. Other treatment possibilities such as gene therapy (if applicable): If relevant, the manuscript should discuss emerging treatment possibilities, such as gene therapy, for diabetic macular edema. This could involve providing an overview of current research or clinical trials investigating the use of gene therapy in the management of DME and discussing its potential benefits and limitations.

5. Improvement of English language: The manuscript would benefit from a thorough proofreading and editing to improve the clarity, grammar, and overall quality of the English language. This includes addressing any typos, grammatical errors, awkward phrasing, or ambiguous statements that may hinder the understanding of the content.

6. Comparison of efficacy and safety: It may be beneficial to discuss more about the comparative efficacy and safety profiles of laser photocoagulation therapy and anti-VEGF therapy in the management of diabetic macular edema. Highlighting the advantages and limitations of each approach can provide a more comprehensive understanding for readers.

7. Cost-effectiveness analysis: Considering the economic impact of laser photocoagulation therapy and its comparison with anti-VEGF therapy in terms of cost-effectiveness can add valuable insights for clinicians and healthcare providers. This aspect should be discussed if data are available in the literature.

8. Future directions: It would be beneficial to conclude the review by discussing potential future directions in the field of diabetic macular edema treatment. This could include mentioning ongoing research, technological advancements, or emerging therapies that may shape the management strategies in the coming years.

The manuscript would benefit from a thorough proofreading and editing to improve the clarity, grammar, and overall quality of the English language. This includes addressing any typos, grammatical errors, awkward phrasing, or ambiguous statements that may hinder the understanding of the content.

Author Response

This manuscript is a resubmission of an earlier submission. The following is a list of the peer review reports and author responses from that submission.

Round 1

Reviewer 1 Report

*The authors aimed to elucidate the role of laser photocoagulation therapy in the treatment of diabetic macular edema (DME). It was a well-written paper. I think a minor revision is required before further consideration of the manuscript.

*Your manuscript should be rechecked for English language, grammar, punctuation, spelling, and overall style. A minor language polishing is required.

*Line 159; “Kato F et al.” should be replaced with “Kato et al.”. Several similar errors are still there (Lines 175, 199, 204, 277, 313, etc.).

*To strengthen your work, I recommend drawing a table summarizing different types of lasers, their properties, advantages, disadvantages, etc.

*Schematic figures of different laser types showing spot size, number of spots, duration, location of action, etc. would be helpful.

*Your manuscript should be rechecked for English language, grammar, punctuation, spelling, and overall style. A minor language polishing is required.

Reviewer 2 Report

It is a well written review

Please consider the following:

1.     Intruduction: please consider quoting EURETINA recommendations for classic laser photocoagulation in DME. (Schmidt-Erfurth U, Garcia-Arumi J, Bandello F, Berg K, Chakravarthy U, Gerendas BS, Jonas J, Larsen M, Tadayoni R, Loewenstein A. Guidelines for the Management of Diabetic Macular Edema by the European Society of Retina Specialists (EURETINA). Ophthalmologica. 2017;237(4):185-222. doi: 10.1159/000458539. Epub 2017 Apr 20. PMID: 28423385.)

2.     SMPL – please refer to treatment of eyes with DME and good visual acuity (Luttrull et al.). Subthreshold micropulse could be a good option for such patients. 

3.     SMPL- please refer to EURETINA guidelines for subthreshold laser use in DME.

4.     Line 216-220 – this sounds too much like a commercial. Please amend. Besides Quantel Medical Supra 577 was also available as far as I remember.

5.     Line 246 – as authors list SRT, the 2-RT laser for the treatment of DME also has to be mentioned (the same idea with targeting melanosomes).  There is not so many papers published, but please include a short paragraph on nano-lasers as well.

6.     Line 355 – please refer to the potential role of angio-OCT in navigated lasers.

7.     Lines 39-391 – I would avoid such statement. Current recommendations are to switch intravitreal medication first. Laser treatment remains still an option for extrafoveal vasogenic DME. Please soften the tone.

English language quality is fine.

Reviewer 3 Report

This is a well - written review. 

Major comments

The authors have nicely summarised the current knowledge and the different types of macular laser in diabetic macular oedema (DMO) treatment. However, the review lacks a critical comparison and detailed discussion on the possible advantages and disadvantages of laser vs. anti-VEGF.  Moreover, the authors didn't mention the steroids, which also have a very important role and have replaced laser and even anti-VEGF to a large extent, particularly in chronic and refractory DMO (Ozurdex, Iluvien). Cost effectiveness and health economics studies (with comparisons between laser vs. anti-VEGF vs. steroid implants) should also be included.

Taken into account the above, I believe that a more detailed discussion on the advantages and disadvantages of macular laser versus anti-VEGF and steroids would improve the manuscript. 

Minor comments  

1) Title should be modified (to reflect also the steroids in the treatment of DMO) as: 

"The role of laser photocoagulation in treating diabetic macular edema in the era of intravitreal drug administration; A Descriptive Review" 

2) More studies should be included, such as

Koushan K, Eshtiaghi A, Fung P, Berger AR, Chow DR. Treatment of Diabetic Macular Edema with Aflibercept and Micropulse Laser (DAM Study). Clin Ophthalmol. 2022 Apr 8;16:1109-1115. doi: 10.2147/OPTH.S360869. PMID: 35422607; PMCID: PMC9005121.

Kanar HS, Arsan A, Altun A, Akı SF, Hacısalihoglu A. Can subthreshold micropulse yellow laser treatment change the anti-vascular endothelial growth factor algorithm in diabetic macular edema? A randomized clinical trial. Indian J Ophthalmol. 2020 Jan;68(1):145-151. doi: 10.4103/ijo.IJO_350_19. PMID: 31856493; PMCID: PMC6951119.

Busch C, Fraser-Bell S, Zur D, Rodríguez-Valdés PJ, Cebeci Z, Lupidi M, Fung AT, Gabrielle PH, Giancipoli E, Chaikitmongkol V, Okada M, Laíns I, Santos AR, Kunavisarut P, Sala-Puigdollers A, Chhablani J, Ozimek M, Hilely A, Unterlauft JD, Loewenstein A, Iglicki M, Rehak M; International Retina Group. Real-world outcomes of observation and treatment in diabetic macular edema with very good visual acuity: the OBTAIN study. Acta Diabetol. 2019 Jul;56(7):777-784. doi: 10.1007/s00592-019-01310-z. Epub 2019 Mar 22. PMID: 30903434; PMCID: PMC6558052.

Moisseiev E, Abbassi S, Thinda S, Yoon J, Yiu G, Morse LS. Subthreshold micropulse laser reduces anti-VEGF injection burden in patients with diabetic macular edema. Eur J Ophthalmol. 2018 Jan;28(1):68-73. doi: 10.5301/ejo.5001000. PMID: 28731494.
